# Identifying Generalization Properties in Neural Networks

## Abstract

While it has not yet been proven, empirical evidence suggests that model generalization is related to local properties of the optima which can be described via the Hessian. We connect model generalization with the local property of a solution under the PAC-Bayes paradigm. In particular, we prove that model generalization ability is related to the Hessian, the higher-order "smoothness" terms characterized by the Lipschitz constant of the Hessian, and the scales of the parameters. Guided by the proof, we propose a metric to score the generalization capability of the model, as well as an algorithm that optimizes the perturbed model accordingly.

## 1 Introduction

Deep models have proven to work well in applications such as computer vision (Krizhevsky et al., 2012) (He et al., 2014) (Karpathy et al., 2014), speech recognition (Mohamed et al., 2012) (Hinton et al., 2012), and natural language processing (Socher et al., 2013) (Graves, 2013) (McCann et al., 2018). Many deep models have millions of parameters, which is more than the number of training samples, but the models still generalize well (Huang et al., 2017).

On the other hand, classical learning theory suggests the model generalization capability is closely related to the "complexity" of the hypothesis space, usually measured in terms of number of parameters, Rademacher complexity or VC-dimension. This seems to be a contradiction to the empirical observations that over-parameterized models generalize well on the test data[1]. Indeed, even if the hypothesis space is complex, the final solution learned from a given training set may still be simple. This suggests the generalization capability of the model is also related to the property of the solution.

Keskar et al. (2017) and Chaudhari et al. (2017) empirically observe that the generalization ability of a model is related to the spectrum of the Hessian matrix $\nabla^2 L(w^*)$ evaluated at the solution, and large eigenvalues of the $\nabla^2 L(w^*)$ often leads to poor model generalization. Also, (Keskar et al., 2017), (Chaudhari et al., 2017) and (Novak et al., 2018b) introduce several different metrics to measure the "sharpness" of the solution, and demonstrate the connection between the sharpness metric and the generalization empirically. Dinh et al. (2017) later points out that most of the Hessian-based sharpness measures are problematic and cannot be applied directly to explain generalization. In particular, they show that the geometry of the parameters in RELU-MLP can be modified drastically by re-parameterization.

Another line of work originates from Bayesian analysis. Mackay (1995) first introduced Taylor expansion to approximate the (log) posterior, and considered the second-order term, characterized by the Hessian of the loss function, as a way of evaluating the model simplicity, or "Occam factor". Recently Smith & Le (2018) use this factor to penalize sharp minima, and determine the optimal batch size. Germain et al. (2016) connect the PAC-Bayes bound and the Bayesian marginal likelihood when the loss is (bounded) negative log-likelihood, which leads to an alternative perspective on Occam's razor. (Langford & Caruana, 2001), and more recently, (Harvey et al., 2017) (Neyshabur et al., 2017) (Neyshabur et al., 2018) use PAC-Bayes bound to analyze the generalization behavior of the deep models.

Since the PAC-Bayes bound holds uniformly for all "posteriors", it also holds for some particular "posterior", for example, the solution parameter perturbed with noise. This provides a natural

---

[1] For example over-parameterized neural network can fit any function of sample size $n$, making the Rademacher complexity large, but empirically those neural networks generalizes well. (Zhang et al., 2016)

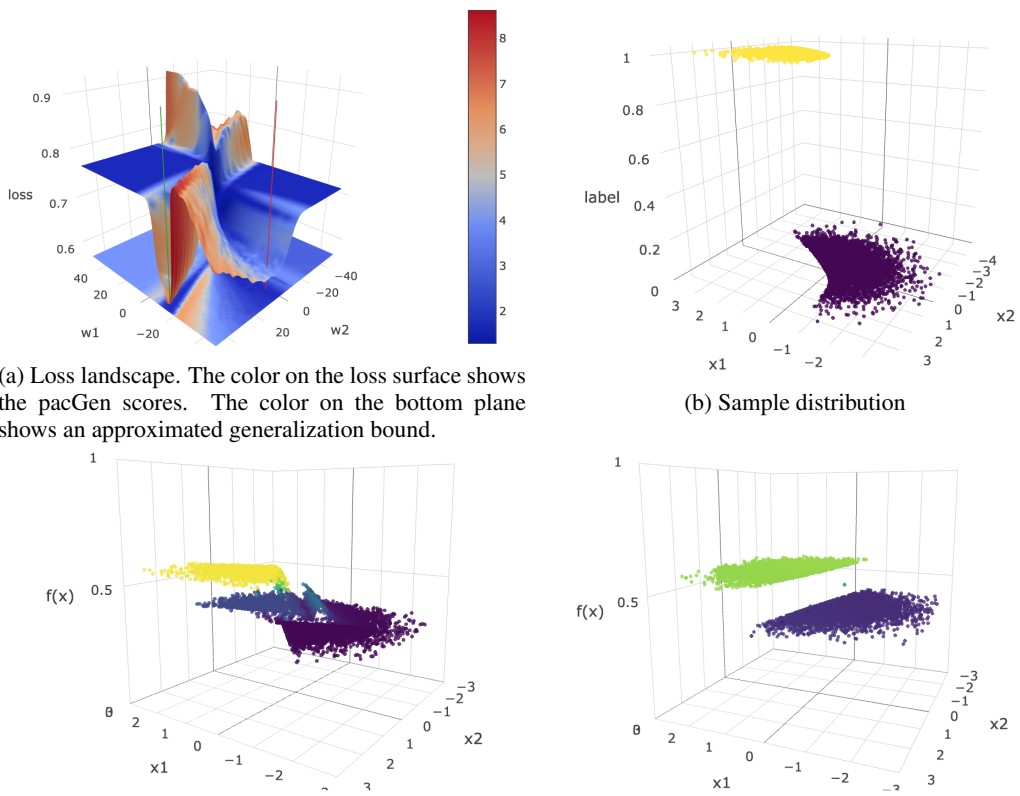

(a) Loss landscape. The color on the loss surface shows the pacGen scores. The color on the bottom plane shows an approximated generalization bound.

(b) Sample distribution

(c) Predicted labels by the sharp minimum

(d) Predicted labels by the flat minimum

Figure 1: Loss Landscape and Predicted Labels of a 5-layer MLP with 2 parameters. The sharp minimum, even though it approximates the true label better, has some complex structures in its predicted labels, while the flat minimum seems to produce a simpler classification boundary.[2]

way to incorporate the local property of the solution into the generalization analysis. In particular, Neyshabur et al. (2017) suggests to use the difference between the perturbed loss and the empirical loss as the sharpness metric. Dziugaite & Roy (2017) tries to optimize the PAC-Bayes bound instead for a better model generalization. Still some fundamental questions remain unanswered. In particular we are interested in the following question:

*How is model generalization related to local "smoothness" of a solution?*

In this paper we try to answer the question from the PAC-Bayes perspective. Under mild assumptions on the Hessian of the loss function, we prove the generalization error of the model is related to this Hessian, the Lipschitz constant of the Hessian, the scales of the parameters, as well as the number of training samples. The analysis also gives rise to a new metric for generalization. Based on this, we can approximately select an optimal perturbation level to aid generalization which interestingly turns out to be related to Hessian as well. Inspired by this observation, we propose a perturbation based algorithm that makes use of the estimation of the Hessian to improve model generalization.

## 2 PAC-BAYES AND MODEL GENERALIZATION

We consider the supervised learning in PAC-Bayes scenario (McAllester, 2003) (McAllester, 1998) (McAllester, 1999) (Langford & Shawe-Taylor, 2002). Suppose we have a labeled data set $\mathcal{S} = \{s_i = (x_i, y_i) \mid i \in \{1, \ldots, n\}, x_i \in \mathbb{R}^d, y_i \in \{0, 1\}^k\}$, where $(x_i, y_i)$ are sampled i.i.d. from a distribution $x_i, y_i \sim \mathfrak{D}_s$.

---

[2]The variables from different layers are shared so that the model only has two free parameters $w_1$ and $w_2$. The bound in (a) is approximated with $\eta = 39$ using inequality (8)

The PAC-Bayes paradigm assumes probability measures over the function class $\mathfrak{F} : \mathcal{X} \to \mathcal{Y}$. In particular, it assumes a "posterior" distribution $\mathfrak{D}_f$ as well as a "prior" distribution $\pi_f$ over the function class $\mathfrak{F}$. We are interested in minimizing the expected loss, in terms of both the random draw of samples as well as the random draw of functions:

$$L(\mathfrak{D}_f, \mathfrak{D}_s) = \mathbb{E}_{f \sim \mathfrak{D}_f} \mathbb{E}_{x,y \sim \mathfrak{D}_s} l(f, x, y).$$

Correspondingly, the empirical loss in the PAC-Bayes paradigm is the expected loss over the draw of functions from the posterior:

$$\hat{L}(\mathfrak{D}_f, \mathcal{S}) = \mathbb{E}_{f \sim \mathfrak{D}_f} \frac{1}{n} \sum_{i=1}^{n} l(f, x_i, y_i) \tag{1}$$

PAC-Bayes theory suggests the gap between the expected loss and the empirical loss is bounded by a term that is related to the KL divergence between $\mathfrak{D}_f$ and $\pi_f$ (McAllester, 1999) (Langford & Shawe-Taylor, 2002). In particular, if the function $f$ is parameterized as $f(w)$ with $w \in \mathcal{W}$, when $\mathfrak{D}_w$ is perturbed around any $w$, we have the following PAC-Bayes bound (Seldin et al., 2012a) (Seldin et al., 2012b) (Neyshabur et al., 2017) (Neyshabur et al., 2018):

**Theorem 1** (PAC-Bayes-Hoeffding Perturbation). *Let $l(f, x, y) \in [0, 1]$, and $\pi$ be any fixed distribution over the parameters $\mathcal{W}$. For any $\delta > 0$ and $\eta > 0$, with probability at least $1 - \delta$ over the draw of $n$ samples, for any $w$ and any random perturbation $u$,*

$$\mathbb{E}_u[L(w + u)] \leq \mathbb{E}_u[\hat{L}(w + u)] + \frac{KL(w + u \| \pi) + \log \frac{1}{\delta}}{\eta} + \frac{\eta}{2n} \tag{2}$$

One may further optimize $\eta$ to get a bound that scales approximately as $\mathbb{E}_u[L(w + u)] \lesssim \mathbb{E}_u[\hat{L}(w + u)] + 2\sqrt{\frac{KL(w+u\|\pi) + \log\frac{1}{\delta}}{2n}}$ (Seldin et al., 2012b).[3] A nice property of the perturbation bound (2) is it connects the generalization with the local properties around the solution $w$ through some perturbation $u$ around $w$. In particular, suppose $\hat{L}(w^*)$ is a local optimum, when the perturbation level of $u$ is small, $\mathbb{E}_u[\hat{L}(w^* + u)]$ tends to be small, but $KL(w^* + u \| \pi)$ may be large since the posterior is too "focused" on a small neighboring area around $w^*$, and vice versa. As a consequence, we may need to search for an "optimal" perturbation level for $u$ so that the bound is minimized.

## 3 MAIN RESULT

While some researchers have already discovered empirically the generalization ability of the models is related to the second order information around the local optima, to the best of our knowledge there is no work on how to connect the Hessian matrix $\nabla^2 \hat{L}(w)$ with the model generalization rigorously. In this section we introduce the local smoothness assumption, as well as our main theorem.

It may be unrealistic to assume global smoothness properties for the deep models. Usually the assumptions only hold in a small local neighborhood $Neigh(w^*)$ around a reference point $w^*$. In this paper we define the neighborhood set as

$$Neigh_\kappa(w^*) = \{w \mid |w_i - w_i^*| \leq \kappa_i \ \forall i\}$$

where $\kappa_i \in \mathbb{R}^+$ is the "radius" of the $i$-th coordinate. In our draft we focus on a particular type of radius $\kappa_i(w^*) = \gamma |w_i^*| + \epsilon$, but our argument holds for other types of radius, too.

In order to get a control of the deviation of the optimal solution we need to assume in $Neigh_{\gamma,\epsilon}(w^*)$, the empirical loss function $\hat{L}$ in (1) is Hessian Lipschitz, which is defined as:

**Definition 1** (Hessian Lipschitz). *A twice differentiable function $f(\cdot)$ is $\rho$-Hessian Lipschitz if:*

$$\forall w_1, w_2, \|\nabla^2 f(w_1) - \nabla^2 f(w_2)\| \leq \rho \|w_1 - w_2\|, \tag{3}$$

where $\| \cdot \|$ is the operator norm.

The Hessian Lipschitz condition has been used in the numeric optimization community to model the smoothness of the second-order gradients (Nesterov & Polyak, 2006) (Carmon et al., 2018) (Jin et al., 2018). In the rest of the draft we always assume the following:

---

[3]Since $\eta$ cannot depend on the data, one has to build a grid and use the union bound.

**Assumption 1.** *In $Neigh_\kappa(w^*)$ the empirical loss $\hat{L}(w)$ defined in (1) is convex, and $\rho$-Hessian Lipschitz.*

For the uniform perturbation, the following theorem holds:

**Theorem 2.** *Suppose the loss function $l(f,x,y) \in [0,1]$, and model weights are bounded $|w_i| + \kappa_i(w) \leq \tau_i \ \forall i$. With probability at least $1 - \delta$ over the draw of $n$ samples, for any $\check{w} \in \mathbb{R}^m$ such that assumption 1 holds*

$$\mathbb{E}_u[L(\check{w} + u)] \leq \hat{L}(\check{w}) + O\left(\sqrt{\frac{m + \sum_i \log \frac{\tau_i}{\check{\sigma}_i} + \log \frac{1}{\delta}}{n}}\right)$$

*where $u_i \sim U(-\check{\sigma}_i, \check{\sigma}_i)$ are i.i.d. uniformly distributed random variables, and*

$$\check{\sigma}_i(\check{w}, \eta, \gamma) = \min\left(\sqrt{\frac{1}{\sqrt{mn}(\nabla_{i,i}^2 \hat{L}(\check{w})/3 + \rho m^{1/2} \kappa_i(\check{w})/9)}}, \kappa_i(\check{w})\right) \qquad (4)$$

Theorem 2 says if we choose the perturbation levels carefully, the expected loss of a uniformly perturbed model is controlled. The bound is related to the diagonal element of Hessian (logarithmic), the Lipschitz constant $\rho$ of the Hessian (logarithmic), the neighborhood scales characterized by $\kappa$ (logarithmic), the number of parameters $m$, and the number of samples $n$. Also roughly the perturbation level is inversely related to $\sqrt{\nabla_{i,i}^2 \hat{L}}$, suggesting the model be perturbed more along the coordinates that are "flat".[4]

Similar argument can be made on the truncated Gaussian perturbation, which is presented in Appendix B. In the next section we walk through some intuitions of our arguments.

## 4 CONNECTING GENERALIZATION AND HESSIAN

Suppose the empirical loss function $\hat{L}(w)$ satisfies the local Hessian Lipschitz condition, then by Lemma 1 in (Nesterov & Polyak, 2006), the perturbation of the function around a fixed point can be bounded by terms up to the third-order,

$$\hat{L}(w + u) \leq \hat{L}(w) + \nabla\hat{L}(w)^T u + \frac{1}{2}u^T \nabla^2 \hat{L}(w)u + \frac{1}{6}\rho\|u\|^3 \ \text{ for } \ w + u \in Neigh_\kappa(w) \qquad (5)$$

For perturbations with zero expectation, i.e., $\mathbb{E}[u] = 0$, the linear term in (5), $\mathbb{E}_u[\nabla\hat{L}(w)^T u] = 0$. Because the perturbation $u_i$ for different parameters are independent, the second order term can also be simplified, since $\mathbb{E}_u[\frac{1}{2}u^T \nabla^2 \hat{L}(w)u] = \frac{1}{2}\sum_i \nabla_{i,i}^2 \hat{L}(w)\mathbb{E}[u_i^2]$.

Considering (2),(5) and assumption 1, it is straightforward to see the bound below holds with probability at least $1 - \delta$

$$\mathbb{E}_u[L(w^* + u)] \leq \hat{L}(w^*) + \frac{1}{2}\sum_i \nabla_{i,i}^2 \hat{L}(w^*)\mathbb{E}[u_i^2] + \frac{\rho}{6}\mathbb{E}[\|u\|^3] + \frac{KL(w^* + u||\pi) + \log\frac{1}{\delta}}{\eta} + \frac{\eta}{2n} \qquad (6)$$

Suppose $u_i \sim U(-\sigma_i, \sigma_i)$, and $\sigma_i \leq \kappa_i(w) \ \forall i$. That is, the "posterior" distribution of the model parameters are uniform distribution, and the distribution supports vary for different parameters. We also assume the perturbed parameters are bounded, i.e., $|w_i| + \kappa_i(w) \leq \tau_i \ \forall i$.[5] If we choose the prior $\pi$ to be $u_i \sim U(-\tau_i, \tau_i)$, and then $KL(w + u||\pi) = \sum_i \log(\tau_i/\sigma_i)$.

The third order term in (6) is bounded by

$$\frac{\rho}{6}\mathbb{E}[\|u\|^3] \leq \frac{\rho m^{1/2}}{6}\mathbb{E}[\|u\|_3^3] \leq \frac{\rho m^{1/2}}{6}\sum_i \kappa_i(w)\mathbb{E}[u_i^2] = \frac{\rho m^{1/2}}{18}\sum_i \kappa_i(w)\sigma_i^2,$$

---

[4]Unfortunately the bound in theorem 2 does not explain the over-parameterization phenomenon since when $m \gg n$ the right hand side explodes.

[5]One may also assume the same $\tau$ for all parameters for a simpler argument. The proof procedure goes through in a similar way.

where we use the inequality $\|u\|_2 \leq m^{\frac{1}{6}}\|u\|_3$ and $m$ is the number of parameters. Pluging in (6), we get

$$\mathbb{E}_u[L(w+u)] \leq \hat{L}(w) + \frac{1}{6}\sum_i \nabla_{i,i}^2 L(w)\sigma_i^2 + \frac{\rho m^{1/2}}{18}\sum_i \kappa_i(w)\sigma_i^2 + \frac{\sum_i \log\frac{\tau_i}{\sigma_i} + \log\frac{1}{\delta}}{\eta} + \frac{\eta}{2n} \tag{7}$$

Solve for $\sigma$ that minimizes the right hand side, and we have the following lemma:

**Lemma 3.** *Suppose the loss function $l(f, x, y) \in [0, 1]$, and model weights are bounded $|w_i| + \kappa_i(w) \leq \tau_i \;\; \forall i$. Given any $\delta > 0$ and $\eta > 0$, with probability at least $1 - \delta$ over the draw of $n$ samples, for any $w^* \in \mathbb{R}^m$ such that assumption 1 holds,*

$$\mathbb{E}_u[L(w^* + u)] \leq \hat{L}(w^*) + \frac{m/2 + \sum_i \log\frac{\tau_i}{\sigma_i^*} + \log\frac{1}{\delta}}{\eta} + \frac{\eta}{2n} \tag{8}$$

*where $u_i \sim U(-\sigma_i^*, \sigma_i^*)$ are i.i.d. uniformly perturbed random variables, and*

$$\sigma_i^*(w^*, \eta, \gamma) = \min\left(\sqrt{\frac{1}{\eta(\nabla_{i,i}^2 L(w^*)/3 + \rho m^{1/2}\kappa_i(w^*)/9)}}, \kappa_i(w^*)\right). \tag{9}$$

In our experiment, we simply treat $\eta$ as a hyper-parameter. Other other hand, one may further build a weighted grid over $\eta$ and optimize for the best $\eta$ (Seldin et al., 2012b). That leads to Theorem 2. Details of the proof are presented in the Appendix C and D.

## 5    On the Re-parameterization of RELU-MLP

Dinh et al. (2017) points out the spectrum of $\nabla^2\hat{L}$ itself is not enough to determine the generalization power. In particular, for a multi-layer perceptron with RELU as the activation function, one may re-parameterize the model and scale the Hessian spectrum arbitrarily without affecting the model prediction and generalization when cross entropy (negative log likelihood) is used as the loss and $w^*$ is the "true" parameter of the sample distribution.

In general our bound does not assume the loss to be the cross entropy. Also we do not assume the model is RELU-MLP. As a result we would not expect our bound stays exactly the same during the re-parameterization. On the other hand, the optimal perturbation levels in our bound scales inversely when the parameters scale, so the bound only changes approximately with a speed of logarithmic factor. According to Lemma (3), if we use the optimal $\sigma^*$ on the right hand side of the bound, $\nabla^2\hat{L}(w)$, $\rho$, and $w^*$ are all behind the logarithmic function. As a consequence, for RELU-MLP, if we do the re-parameterization trick, the change of the bound is small.

In the next two sections we introduce some heuristic-based approximations enlightened by the bound, as well as some interesting empirical observations.

## 6    An Approximate Generalization Metric

Assuming $\hat{L}(w)$ is locally convex around $w^*$, so that $\nabla_{i,i}^2\hat{L}(w^*) \geq 0$ for all $i$. If we look at Lemma 3, for fixed $m$ and $n$, the only relevant term is $\sum_i \log\frac{\tau_i}{\sigma_i^*}$. Replacing the optimal $\sigma^*$, and using $|w_i| + \kappa_i(w)$ to approximate $\tau_i$, we come up with **PAC**-Bayes based **Gen**eralization metric, called pacGen,[6]

$$\Psi_\kappa(\hat{L}, w^*) = \sum_i \log\left((|w_i^*| + \kappa_i(w^*))\max\left(\sqrt{\nabla_{i,i}^2\hat{L}(w^*) + \rho(w^*)\sqrt{m}\kappa_i(w^*)}, \frac{1}{\kappa_i(w^*)}\right)\right).$$

---

[6] Even though we assume the local convexity in our metric, in application we may calculate the metric on every points. When $\nabla_{i,i}^2\hat{L}(w^*) + \rho(w^*)\sqrt{m}\kappa_i(w^*) < 0$ we simply treat it as 0.

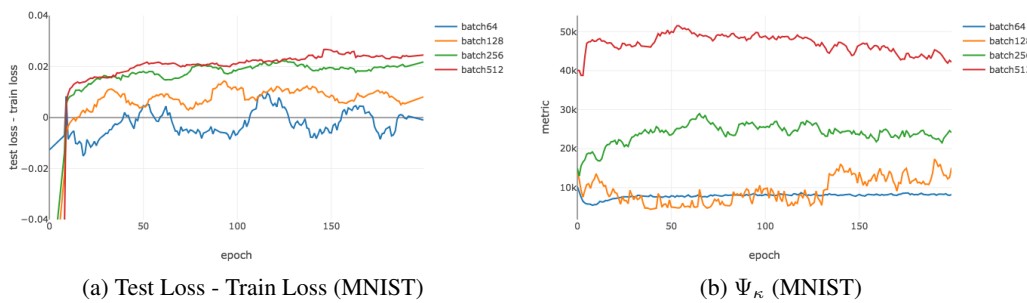

(a) Test Loss - Train Loss (MNIST)        (b) $\Psi_\kappa$ (MNIST)

Figure 2: Generalization gap and $\Psi_\kappa$ as a function of epochs on MNIST for different batch sizes.

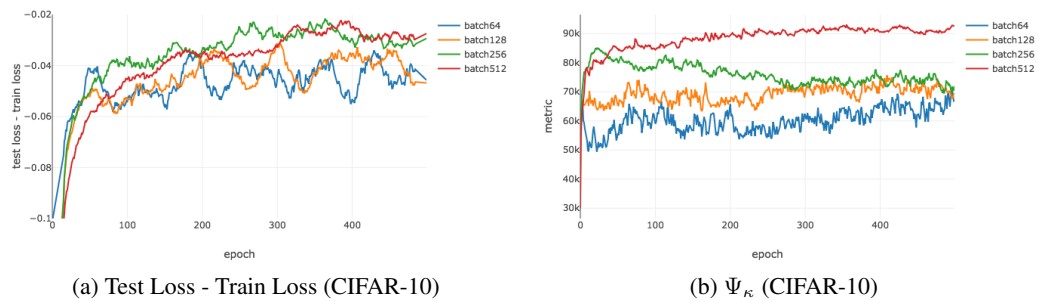

(a) Test Loss - Train Loss (CIFAR-10)        (b) $\Psi_\kappa$ (CIFAR-10)

Figure 3: Generalization gap and $\Psi_\kappa$ as a function of epochs on CIFAR-10 for different batch sizes. SGD is used as the optimizer, and the learning rate is set as $0.01$ for all configurations.

A self-explained toy example is displayed in Figure 1. To calculate the metric on real-world data we need to estimate the diagonal elements of the Hessian $\nabla^2 \hat{L}$ as well as the Lipschitz constant $\rho$ of the Hessian. For efficiency concern we follow Adam (Kingma & Ba, 2014) and approximate $\nabla^2_{i,i} \hat{L}$ by $(\nabla \hat{L}[i])^2$. Also we use the exponential smoothing technique with $\beta = 0.999$ as in (Kingma & Ba, 2014).

To estimate $\rho$, we first estimate the Hessian of a randomly perturbed model $\nabla^2 \hat{L}(w + u)$, and then approximate $\rho$ by $\rho = \max_i \frac{|\nabla^2_i L(w+u_i) - \nabla^2_i L(w)|}{|u_i|}$. For the neighborhood radius $\kappa$ we use $\gamma = 0.1$ and $\epsilon = 0.1$ for all the experiments in this section.

We used the same model without dropout from the PyTorch example [7]. Fixing the learning rate as $0.1$, we vary the batch size for training. The gap between the test loss and the training loss, and the metric $\Psi_\kappa(\hat{L}, w^*)$ are plotted in Figure 2. We had the same observation as in (Keskar et al., 2017) that as the batch size grows, the gap between the test loss and the training loss tends to get larger. Our proposed metric $\Psi_\kappa(\hat{L}, w^*)$ also shows the exact same trend. Note we do not use LR annealing heuristics as in (Goyal et al., 2017) which enables large batch training.

Similarly we also carry out experiment by fixing the training batch size as $256$, and varying the learning rate. Figure 4 shows generalization gap and $\Psi_\kappa(\hat{L}, w^*)$ as a function of epochs. It is observed that as the learning rate decreases, the gap between the test loss and the training loss increases. And the proposed metric $\Psi_\kappa(\hat{L}, w^*)$ shows similar trend compared to the actual generalization gap. Similar trends can be observed if we run the same model on CIFAR-10 (Krizhevsky, 2009) as shown in Figure 3 and Figure 5.

## 7   A PERTURBED OPTIMIZATION ALGORITHM

Adding noise to the model for better generalization has proven successful both empirically and theoretically (Zhu et al., 2018) (Hoffer et al., 2017) (Jastrzębski et al., 2017) (Dziugaite & Roy, 2017) (Novak et al., 2018a). Instead of only minimizing the empirical loss, (Langford & Caruana,

---
[7]https://github.com/pytorch/examples/tree/master/mnist

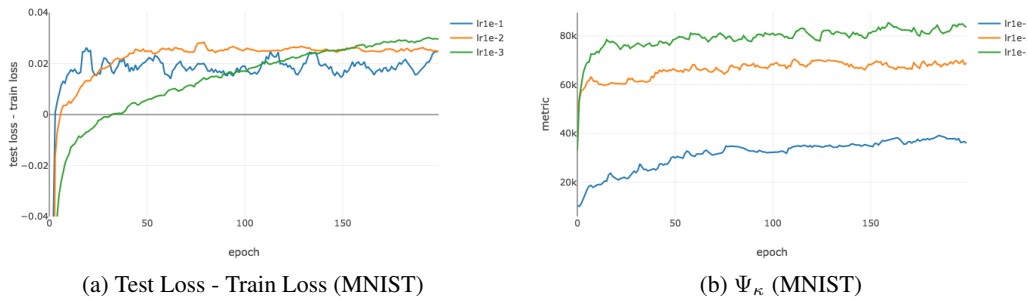

(a) Test Loss - Train Loss (MNIST)  (b) $\Psi_\kappa$ (MNIST)

Figure 4: Generalization gap and $\Psi_\kappa$ as a function of epochs on MNIST for different learning rates. SGD is used as the optimizer, and the batch size is set as 256 for all configurations. As the learning rate shrinks, $\Psi_\kappa(\hat{L}, w^*)$ gets larger. The trend is consistent with the true gap of losses.

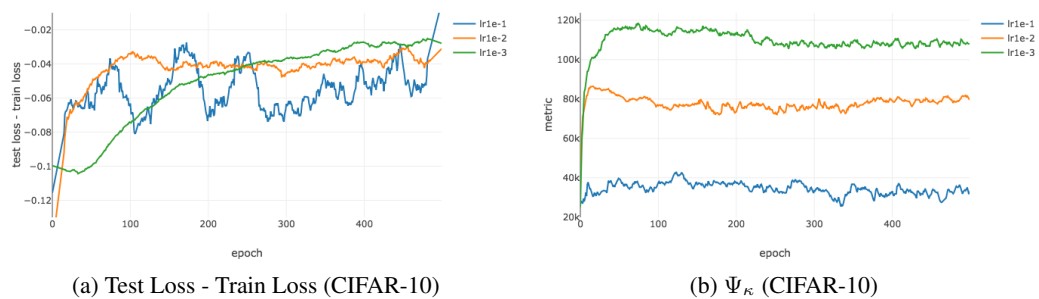

(a) Test Loss - Train Loss (CIFAR-10)  (b) $\Psi_\kappa$ (CIFAR-10)

Figure 5: Generalization gap and $\Psi_\kappa$ as a function of epochs on CIFAR-10 for different learning rates. SGD is used as the optimizer, and the batch size is set as 256 for all configurations.

2001) and (Dziugaite & Roy, 2017) assume different perturbation levels on different parameters, and minimize the generalization bound led by PAC-Bayes for better model generalization. However how to integrate the smoothness property of the local optima is not clear.

The right hand side of (2) has $\mathbb{E}_u[\hat{L}(w + u)]$. This suggests rather than minimizing the empirical loss $\hat{L}(w)$, we should optimize the perturbed empirical loss $\mathbb{E}_u[\hat{L}(w + u)]$ instead for a better model generalization power.

We introduce a systematic way to perturb the model weights based on the PAC-Bayes bound. Again we use the same exponential smoothing technique as in Adam (Kingma & Ba, 2014) to estimate the Hessian $\nabla^2 \hat{L}$. The details of the algorithm is presented in Algorithm 1, where we treat $\eta$ as a hyper-parameter.

Even though in theoretical analysis $E_u[\nabla \hat{L} \cdot u] = 0$, in applications, $\nabla \hat{L} \cdot u$ won't be zero especially when we only implement 1 trial of perturbation. On the other hand, if the gradient $\nabla \hat{L}$ is close to zero, then the first order term can be ignored. As a consequence, in Algorithm 1 we only perturb the parameters that have small gradients whose absolute value is below $\beta_2$. For efficiency issues we used a per-parameter $\rho_i$ capturing the variation of the diagonal element of Hessian. Also we decrease the perturbation level with a log factor as the epoch increases.

We compare the perturbed algorithm against the original optimization method on CIFAR-10, CIFAR-100 (Krizhevsky, 2009), and Tiny ImageNet[8]. The results are shown in Figure 6. We use the Wide-ResNet (Zagoruyko & Komodakis, 2018) as the prediction model.[9] The depth of the chosen model is 58, and the widen-factor is set as 3. The dropout layers are turned off. For CIFAR-10 and CIFAR-100, we use Adam with a learning rate of $10^{-4}$, and the batch size is 128. For the perturbation parameters we use $\eta = 0.01$, $\gamma = 10$, and $\epsilon$=1e-5. For Tiny ImageNet, we use SGD with learning rate $10^{-2}$, and the batch size is 200. For the perturbed SGD we set $\eta = 100$, $\gamma = 1$,

---

[8] https://tiny-imagenet.herokuapp.com/
[9] https://github.com/meliketoy/wide-resnet.pytorch/blob/master/networks/wide_resnet.py

---

**Algorithm 1** Perturbed OPT

---

**Require:** $\eta$, $\gamma = 0.1$, $\beta_1 = 0.999$, $\beta_2 = 0.1$, $\epsilon$=1e-5.

1: Initialization: $\sigma_i \leftarrow 0$ for all $i$. $t \leftarrow 0$, $h_0 \leftarrow 0$
2: **for** epoch in $1, \ldots, N$ **do**
3:     **for** minibatch in one epoch **do**
4:         **for** all $i$ **do**
5:             **if** $t > 0$ **then**
6:                 $\rho[i] \leftarrow \frac{|h_t[i] - h_{t-1}[i]|}{\|w_t - w_{t-1}\|}$
7:                 $\kappa[i] \leftarrow \frac{\gamma}{\log(1+epoch)}|w_{t-1}[i]| + \epsilon$
8:                 $\sigma_i \leftarrow \min\left(\frac{1}{\log(1+epoch)\sqrt{\eta(h_t[i]+\rho[i]\cdot\kappa[i])}}, \kappa[i]\right) \cdot \mathbf{1}_{|g_t[i]|<\beta_2}$
9:             $u_t[i] \sim U(-\sigma_i, \sigma_i)$(sample a set of perturbations)
10:            $g_{t+1} \leftarrow \nabla_w \hat{L}_t(w_t + u_t)$ (get stochastic gradients w.r.t. perturbed loss)
11:            $h_{t+1} \leftarrow \beta_1 h_t + (1 - \beta_1)g_{t+1}^2$ (update second moment estimate)
12:            $w_{t+1} \leftarrow \text{OPT}(w_t)$ (update $w$ using off-the-shell algorithms)
13:            $t \leftarrow t + 1$

---

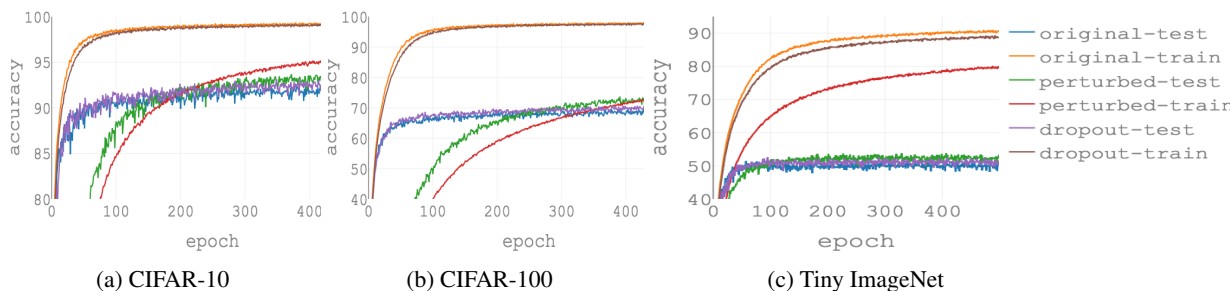

(a) CIFAR-10         (b) CIFAR-100         (c) Tiny ImageNet

Figure 6: Training and testing accuracy as a function of epochs on CIFAR-10, CIFAR-100 and Tiny ImageNet. For CIFAR, Adam is used as the optimizer, and the learning rate is set as $10^{-4}$. For the Tiny ImageNet, SGD is used as the optimizer, and the learning rate is set as $10^{-2}$. The dropout method in the comparison uses $0.1$ as the dropout rate. Details can be found in Appendix G.

and $\epsilon$=1e-5. Also we use the validation set as the test set for the Tiny ImageNet. We observe the effect with perturbation appears similar to regularization. With the perturbation, the accuracy on the training set tends to decrease, but the test on the validation set increases. The perturbedOPT also works better than dropout possibly due to the fact that the it puts different levels of perturbation on different parameters according to the local smoothness structures, while only one dropout rate is set for the all the parameters across the model for the dropout method.

## 8 CONCLUSION

We connect the smoothness of the solution with the model generalization in the PAC-Bayes framework. We prove that the generalization power of a model is related to the Hessian and the smoothness of the solution, the scales of the parameters, as well as the number of training samples. In particular, we prove that the best perturbation level scales roughly as the inverse of the square root of the Hessian, which mostly cancels out scaling effect in the re-parameterization suggested by (Dinh et al., 2017). To the best of our knowledge, this is the first work that integrate Hessian in the model generalization bound rigorously. It also roughly explains the effect of re-parameterization over the generalization. Based on our generalization bound, we propose a new metric to test the model generalization and a new perturbation algorithm that adjusts the perturbation levels according to the Hessian. Finally, we empirically demonstrate the effect of our algorithm is similar to a regularizer in its ability to attain better performance on unseen data.

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

## A  DETAILS OF THE TOY EXAMPLE

This section discusses the details of the toy example shown in Figure (1). We construct a small 2-dimensional sample set from a mixture of $3$ Gaussians, and then binarize the labels by thresholding them from the median value. The sample distribution is shown in Figure 1b. For the model we use a $5$-layer MLP with sigmoid as the activation and cross entropy as the loss. There are no bias terms in the linear layers, and the weights are shared. For the shared 2-by-2 linear coefficient matrix, we treat two entries as constants and optimize the other 2 entries. In this way the whole model has only two free parameters $w_1$ and $w_2$.

The model is trained using 100 samples. Fixing the samples, we plot the loss function with respect to the model variables $\hat{L}(w_1, w_2)$, as shown in Figure 1a. Many local optima are observed even in this simple two-dimensional toy example. In particular: a sharp one, marked by the vertical green line, and a flat one, marked by the vertical red line. The colors on the loss surface display the values of the generalization metric scores (pacGen) defined in Section 6. Smaller metric value indicates better generalization power.

As displayed in the figure, the metric score around the global optimum, indicated by the vertical green bar, is high, suggesting possible poor generalization capability as compared to the local optimum indicated by the red bar. We also plot a plane on the bottom of the figure. The color projected on the bottom plane indicates an approximated generalization bound, which considers both the loss and the generalization metric.[10] The local optimum indicated by the red bar, though has a slightly higher loss, has a similar overall bound compared to the "sharp" global optimum.

On the other hand, fixing the parameter $w_1$ and $w_2$, we may also plot the labels predicted by the model given the samples. Here we plot the prediction from both the sharp minimum (Figure 1c) and the flat minimum (Figure 1d). The sharp minimum, even though it approximates the true label better, has some complex structures in its predicted labels, while the flat minimum seems to produce a simpler classification boundary.

## B  TRUNCATED GAUSSIAN

Because the Gaussian distribution is not bounded but the inequality (5) requires bounded perturbation, we first truncate the distribution. The procedure of truncation is similar to the proof in (Neyshabur et al., 2018) and (McAllester, 2003).

Let $u_i \sim N(0, \sigma_i^2)$. Denote the truncated Gaussian as $N_{\kappa_i}(0, \sigma_i^2)$. If $\tilde{u}_i \sim N_{\kappa_i}(0, \sigma_i^2)$ then

$$\mathbb{P}_{\kappa_i}(\tilde{u}) = \frac{1}{Z_i} \begin{cases} p(u_i) & \text{if } |u_i| < \kappa_i(w) \\ 0 & o.w. \end{cases} \tag{10}$$

Now let's look at the event

$$\mathbf{E} = \{ u \mid |u_i| < \kappa_i(w) \ \forall \ i \} \tag{11}$$

If $\forall i \ \sigma_i < \frac{\kappa_i(w)}{\sqrt{2}\mathrm{erf}^{-1}(\frac{1}{2m})}$, by union bound $\mathbb{P}(\mathbf{E}) \geq 1/2$. Here $\mathrm{erf}^{-1}$ is the inverse Gaussian error function defined as $\mathrm{erf}(x) = \frac{2}{\sqrt{\pi}} \int_0^x e^{-t^2} dt$, and $m$ is the number of parameters. Following a similar procedure as in the proof of Lemma 1 in (Neyshabur et al., 2018),

$$KL(w + \tilde{u} \| \pi) \leq 2(KL(w + u \| \pi) + 1) \tag{12}$$

Suppose the coefficients are bounded such that $\sum_i w_i^2 \leq \tau$, where $\tau$ is a constant. Choose the prior $\pi$ as $N(0, \tau I)$, and we have

$$KL(w + u \| \pi) \leq \frac{1}{2}(m \log \tau - \sum_i \log \sigma_i^2 - m + \frac{1}{\tau} \sum_i \sigma_i^2 + 1) \tag{13}$$

---

[10] the bound was approximated with $\eta = 39$ using inequality (8)

Notice that after the truncation the variance only becomes smaller, so the bound of (6) for the truncated Gaussian becomes

$$\mathbb{E}_u[L(w + \tilde{u})] \leq \hat{L}(w) + \frac{1}{2} \sum_i \nabla_{i,i}^2 L(w) \sigma_i^2 + \frac{\rho m^{1/2}}{6} \sum_i \kappa_i(w) \sigma_i^2$$
$$+ \frac{m \log \tau - \sum_i \log \sigma_i^2 - m + \frac{1}{\tau} \sum_i \sigma_i^2 + 1 + 2 \log \frac{1}{\delta}}{2\eta} + \frac{\eta}{2n} \quad (14)$$

Again when $\hat{L}(w)$ is convex around $w^*$ such that $\nabla^2 \hat{L}(w^*) \geq 0$, solve for the best $\sigma_i$ and we get the following lemma:

**Lemma 4.** *Suppose the loss function $l(f, x, y) \in [0, 1]$, and model weights are bounded $\sum_i w_i^2 \leq \tau$. For any $\delta > 0$ and $\eta$, with probability at least $1 - \delta$ over the draw of $n$ samples, for any $w^* \in \mathbb{R}^m$ such that assumption 1 holds,*

$$\mathbb{E}_u[L(w^* + \tilde{u})] \leq \hat{L}(w^*) + \frac{m \log \tau - \sum_i \log \sigma_i^2 + 1 + 2 \log \frac{1}{\delta}}{2\eta} + \frac{\eta}{2n} \quad (15)$$

*where $\tilde{u}_i \sim N_{\kappa_i}(0, \sigma_i^*)$ are i.i.d. random variables distributed as truncated Gaussian,*

$$\sigma_i^* = \min \left( \sqrt{\frac{1}{\eta \nabla_{i,i}^2 \hat{L}(w^*) + \frac{\rho \eta m^{1/2}}{3} \kappa_i(w^*) + \frac{1}{\tau}}}, \frac{\kappa_i(w^*)}{\sqrt{2} \text{erf}^{-1}(\frac{1}{2m})} \right) \quad (16)$$

*and $\sigma_i^{*2}$ is the $i$-th diagonal element in $\Sigma^*$.*

Again we have an extra term $\eta$, which may be further optimized over a grid to get a tighter bound. In our algorithm we treat $\eta$ as a hyper-parameter instead.

## C  PROOF OF LEMMA 3

*Proof.* We rewrite the inequality (7) below

$$\mathbb{E}_u[L(w + u)] \leq \hat{L}(w) + \frac{1}{6} \sum_i \nabla_{i,i}^2 L(w) \sigma_i^2 + \frac{\rho m^{1/2}}{18} \sum_i \kappa_i(w) \sigma_i^2 + \frac{\sum_i \log \frac{\tau_i}{\sigma_i} + \log \frac{1}{\delta}}{\eta} + \frac{\eta}{2n} \quad (17)$$

The terms related to $\sigma_i$ on the right hand side of (17) are

$$\frac{1}{6} \nabla_{i,i}^2 L(w) \sigma_i^2 + \frac{\rho m^{1/2}}{18} \kappa_i(w) \sigma_i^2 - \frac{\log \sigma_i}{\eta} \quad (18)$$

Since the assumption is $\nabla_{i,i}^2 \hat{L}(w^*) \geq 0$ for all $i$, $\nabla_{i,i}^2 \hat{L}(w) + \rho m^{1/2} \kappa_i(w)/3 > 0$. Solving for $\sigma$ that minimizes the right hand side of (17), and we have

$$\sigma_i^*(w, \eta, \gamma) = \min \left( \sqrt{\frac{1}{\eta(\nabla_{i,i}^2 \hat{L}(w)/3 + \rho m^{1/2} \kappa_i(w)/9)}}, \kappa_i(w) \right) \quad (19)$$

The term $\frac{1}{6} \sum_i \nabla_{i,i}^2 L(w) \sigma_i^2 + \frac{\rho m^{1/2}}{18} \sum_i \kappa_i(w) \sigma_i^2$ on the right hand side of (7) is monotonically increasing w.r.t. $\sigma^2$, so

$$\frac{1}{6} \sum_i \nabla_{i,i}^2 L(w) \sigma_i^{*2} + \frac{\rho m^{1/2}}{18} \sum_i \kappa_i(w) \sigma_i^{*2}$$
$$\leq \sum_i \left( \frac{1}{6} \nabla_{i,i}^2 L(w) + \frac{\rho m^{1/2}}{18} \kappa_i(w) \right) \frac{1}{\eta(\nabla_{i,i}^2 \hat{L}(w)/3 + \rho m^{1/2} \kappa_i(w)/9)}$$
$$= \frac{m}{2\eta} \quad (20)$$

Combine the inequality (20), and the equation (19) with (17), and we complete the proof.

$\square$

## D  PROOF OF THEOREM 2

*Proof.*  Combining (4) and (7), we get

$$\mathbb{E}_u[L(\check{w}+u)] \leq \hat{L}(\check{w}) + \frac{1}{2}\sqrt{\frac{m}{n}} + \frac{\sum_i \log\frac{\tau_i}{\check{\sigma}_i} + \log\frac{1}{\delta}}{\eta} + \frac{\eta}{2n}$$

The following proof is similar to the proof of Theorem 6 in (Seldin et al., 2012b). Note the $\eta$ in Lemma (3) cannot depend on the data. In order to optimize $\eta$ we need to build a grid of the form

$$\eta_j = e^j\sqrt{2n\log\frac{1}{\delta_j}}$$

for $j \geq 0$.

For a given value of $\sum_i \log\frac{\tau_i}{\check{\sigma}_i}$, we pick $\eta_j$, such that

$$j = \left\lfloor \frac{1}{2}\log\left(\frac{\sum_i \log\frac{\tau_i}{\check{\sigma}_i}}{\log\frac{1}{\delta_j}} + 1\right)\right\rceil$$

where $\lfloor x \rceil$ is the largest integer value smaller than $x$. Set $\delta_j = \delta 2^{-(j+1)}$, and take a weighted union bound over $\eta_j$-s with weights $2^{-(j+1)}$, and we have with probability at least $1 - \delta$,

$$\mathbb{E}_u[L(\check{w}+u)] \leq \hat{L}(\check{w}) + \frac{1}{2}\sqrt{\frac{m}{n}} + (1+1/e)\sqrt{\frac{\sum_i \log\frac{\tau_i}{\check{\sigma}_i} + \log\frac{1}{\delta} + \frac{\log 2}{2}\left(2 + \log\left(\frac{\sum_i \log\frac{\tau_i}{\check{\sigma}_i}}{\log\frac{1}{\delta}} + 1\right)\right)}{2n}}$$

Simplify the right hand side and we complete the proof.

$\square$

## E  PROOF OF LEMMA 4

*Proof.*  We first rewrite the inequality (14) below:

$$\mathbb{E}_u[L(w^*+\tilde{u})] \leq \hat{L}(w^*) + \frac{1}{2}\sum_i \nabla^2_{i,i}L(w^*)\sigma_i^2 + \frac{\rho m^{1/2}}{6}\sum_i \kappa_i(w^*)\sigma_i^2$$

$$+ \frac{m\log\tau - \sum_i \log\sigma_i^2 - m + \frac{1}{\tau}\sum_i \sigma_i^2 + 1 + 2\log\frac{1}{\delta}}{2\eta} + \frac{\eta}{2n}$$

The terms related to $\sigma_i$ on the right hand side of (14) is

$$\left(\frac{1}{2}\nabla^2_{i,i}L(w^*) + \frac{\rho m^{1/2}}{6}\kappa_i(w^*) + \frac{1}{2\tau\eta}\right)\sigma_i^2 - \frac{\log\sigma_i^2}{2\eta} \tag{21}$$

Take gradients w.r.t. $\sigma_i$, when $\nabla^2_i\hat{L} \geq 0$, we get the optimal $\sigma_i^*$,

$$\sigma_i^* = \min\left(\sqrt{\frac{1}{\eta\nabla^2_{i,i}\hat{L}(w^*) + \frac{\rho\eta m^{1/2}}{3}\kappa_i(w^*) + \frac{1}{\tau}}}, \frac{\kappa_i(w^*)}{\sqrt{2}\text{erf}^{-1}(\frac{1}{2m})}\right)$$

Note the first term in (21) is monotonously increasing w.r.t. $\sigma_i$, so

$$\left(\frac{1}{2}\nabla^2_{i,i}L(w^*) + \frac{\rho m^{1/2}}{6}\kappa_i(w^*) + \frac{1}{2\tau\eta}\right)\sigma_i^{*2}$$

$$\leq \left(\frac{1}{2}\nabla^2_i L(w^*) + \frac{\rho m^{1/2}}{6}\kappa_i(w^*) + \frac{1}{2\tau\eta}\right)\frac{1}{\eta\nabla^2_i\hat{L}(w^*) + \frac{\rho\eta m^{1/2}}{3}\kappa_i(w^*) + \frac{1}{\tau}}$$

$$= \frac{1}{2\eta} \tag{22}$$

Summing over $m$ parameters and combine (14), we complete the proof.

$\square$

## F    A Lemma about Eigenvalues of Hessian and Generalization

By extrema of the Rayleigh quotient, the quadratic term on the right hand side of inequality (5) is further bounded by

$$u^T \nabla^2 \hat{L}(w)u \leq \lambda_{max}(\nabla^2 \hat{L}(w))\|u\|^2. \tag{23}$$

This is consistent with the empirical observations of Keskar et al. (2017) that the generalization ability of the model is related to the eigenvalues of $\nabla^2 \hat{L}(w)$. The inequality (23) still holds even if the perturbations $u_i$ and $u_j$ are correlated. We add another lemma about correlated perturbations below.

**Lemma 5.** *Suppose the loss function $l(f, x, y) \in [0, 1]$. Let $\pi$ be any distribution on the parameters that is independent from the data. Given $\delta > 0$ $\eta > 0$, with probability at least $1 - \delta$ over the draw of $n$ samples, for any local optimal $w^*$ such that $\nabla \hat{L}(w^*) = 0$, $\hat{L}(w)$ satisfies the local $\rho$-Hessian Lipschitz condition in $Neigh_\kappa(w^*)$, and any random perturbation $u$, s.t., $|u_i| \leq \kappa_i(w^*)$ $\forall i$, we have*

$$\mathbb{E}_u[L(w^* + u)] \leq \hat{L}(w^*) + \frac{1}{2}\lambda_{max}\left(\nabla^2 \hat{L}(w^*)\right)\sum_i \mathbb{E}[u_i^2] + \frac{\rho}{6}\mathbb{E}[\|u\|^3]$$

$$+ \frac{KL(w^* + u||\pi) + \log\frac{1}{\delta}}{\eta} + \frac{\eta}{2n}. \tag{24}$$

*Proof.* The proof of the Lemma 5 is straightforward. Since $\nabla \hat{L}(w^*) = 0$, the first order term is zero at the local optimal point even if $\mathbb{E}[u] \neq 0$. By extrema of the Rayleigh quotient, the quadratic term on the right hand side of inequality (5) is further bounded by

$$u^T \nabla^2 \hat{L}(w)u \leq \lambda_{max}\left(\nabla^2 \hat{L}(w)\right)\|u\|^2. \tag{25}$$

Due to the linearity of the expected value,

$$\mathbb{E}[u^T \nabla^2 \hat{L}(w)u] \leq \lambda_{max}\left(\nabla^2 \hat{L}(w)\right)\sum_i \mathbb{E}[u_i^2], \tag{26}$$

which does not assume independence among the perturbations $u_i$ and $u_j$ for $i \neq j$.

$\square$

## G    PertOPT v.s. Dropout: An Empirical Comparison

This section contains several figures comparing dropout and the proposed perturbation algorithm. Dropout can be viewed as multiplicative perturbation using Bernoulli distribution. It has already been widely used in almost every deep models. For comparison we present results using the exact same wide resnet architectures except the dropout layers are turned on or off. We report the accuracy with dropout rate of 0.0, 0.1, 0.3, and 0.5 on CIFAR-10 and CIFAR-100. For Tiny ImageNet we report the result with dropout rate being 0.0, 0.1, and 0.3. Again for the pertOPT algorithm all the dropout layers are turned off.

The depth of the chosen wide resnet model (Zagoruyko & Komodakis, 2018) is 58, and the widen-factor is set as 3. For CIFAR-10 and CIFAR-100, we use Adam with a learning rate of $10^{-4}$, and the batch size is 128. For the perturbation parameters we use $\eta = 0.01$, $\gamma = 10$, and $\epsilon$=1e-5. For Tiny ImageNet, we use SGD with learning rate $10^{-2}$, and the batch size is 200. For the perturbed SGD we set $\eta = 100$, $\gamma = 1$, and $\epsilon$=1e-5. Also we use the validation set as the test set for the Tiny ImageNet.

Figure (7), (8), and (9) show the accuracy versus epochs for training and validation in CIFAR-10, CIFAR-100, and Tiny ImageNet respectively. It is pretty clear that with added dropout the validation/test accuracy got boosted compared to the original method. For CIFAR-10, dropout rate 0.3 seems to work best compared to all the other dropout configurations. For CIFAR-100 and Tiny

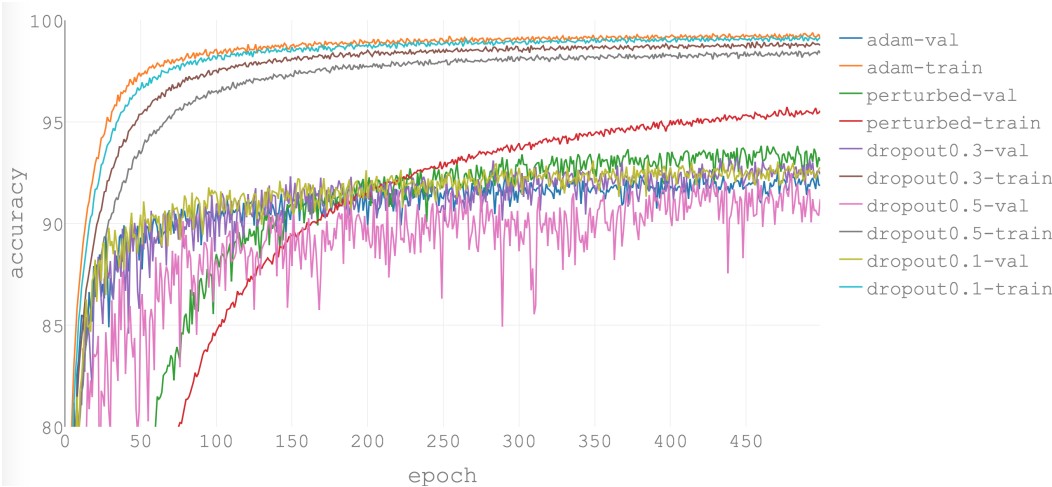

Figure 7: Training and validation accuracy of PertOPT and Dropout on CIFAR-10. Adam-train, and adam-val use the wide resnet model with 0 dropout rate. Perturbed-val and perturbed-train use the same wide resnet with 0 dropout rate, but added perturbation according to algorithm 1.

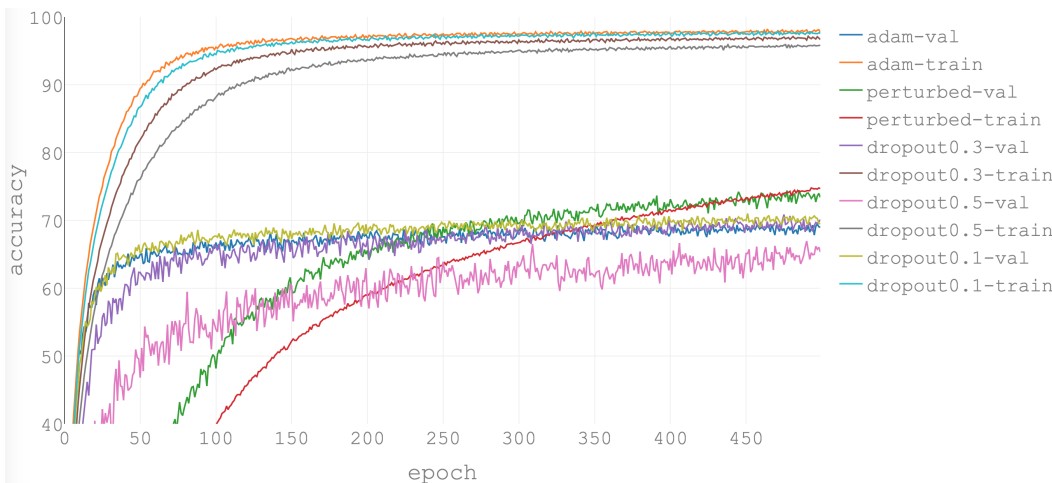

Figure 8: Training and validation accuracy of PertOPT and Dropout on CIFAR-100. Adam-train, and adam-val use the wide resnet model with 0 dropout rate. Perturbed-val and perturbed-train use the same wide resnet with 0 dropout rate, but added perturbation according to algorithm 1.

ImageNet, dropout 0.1 seems to work better. This may be due to the fact that CIFAR-10 has less training samples so more regularization is needed to prevent overfit.

Although both perturbedOPT and dropout can be viewed as certain kind of regularization, in all experiments the perturbed algorithm shows better performance on the validation/test data sets compared to the dropout methods. One possible explanation is maybe the perturbed algorithm puts different levels of perturbation on different parameters according to the local smoothness structures, while only one dropout rate is set for all the parameters across the model.

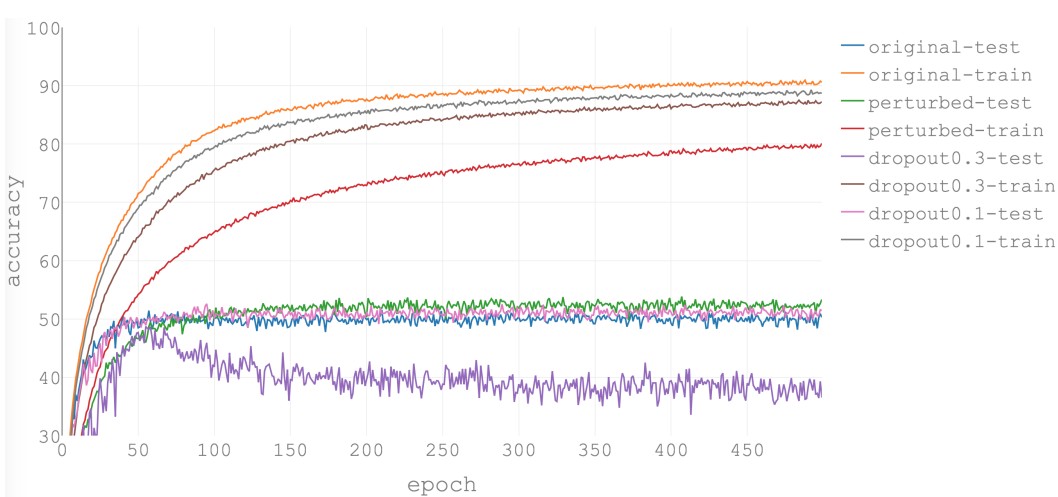

Figure 9: Training and validation accuracy of PertOPT and Dropout on Tiny ImageNet. Original-train, and original-test use the wide resnet model with 0 dropout rate. Perturbed-test and perturbed-train use the same wide resnet with 0 dropout rate, but added perturbation according to algorithm 1.

