# OpenReview forum: "Identifying Generalization Properties in Neural Networks"
_ICLR.cc/2019/Conference_

### Official Review · AnonReviewer1 · 2018-10-22
**The paper does some extensive calculations but is weak on qualitative insights and empirical evaluation.**

**Rating:** 6
**Confidence:** 4

**Review:**

This paper gives various PAC-Bayesian generalization guarantees and some
empirical results on parameter perturbation in training using an algorithm
motivated by the theory.

The fundamental issue addressed in this paper is whether parameter
perturbation during training improves generalization and, if so, what
theoretical basis exists for this phenomenon.  For continuously
parameterized models, PAC-Bayesian bounds are fundamentally based on
parameter perturbation (non-singular posteriors).  So PAC-Bayesian
theory is naturally tied to parameter perturbation issues.  A more
refined question is whether the size of the perturbation should be
done on a per-parameter bases and whether per-parameter noise levels
should be adaptive --- should the appropriate noise level for each
parameter be adjusted on the basis of statistics in the training data.
Adam and RMS-prop both adapt per-parameter learning rate eta_i to be
proportional to 1/((E g_i^2) + epsilon) where E g_i^2 is some running
estimate of the expectation over a draw of a training point of the
square of the gradient of the loss with respect to parameter i.  At
the end of the day, this paper, based on PAC-Bayesian analysis,
proposes that a very similar adaptation be made to per-parameter noise
during training but where E g_i^2 is replaced by the RMS value \sqrt{E
g_i^2}.  It seems that all theoretical analyses require the square
root --- the units need to work.  A fundamental theoretical question,
perhaps unrelated to this paper, is why in learning rate adaptation the
square root hurts the performance.

This paper can be evaluated on both theoretical and empirical grounds.
At a theoretical level I have several complaints.  First, the
theoretical analysis seem fairly mechanical and without theoretical
innovation. Second, the analysis obscures the prior being used (the
learning bias). The paper first states an assumption that each
parameter is a-priori taken to be uniform over |w_i| <= \tau_i and the
KL-divergence in the PAC-Bayes bound is then log tau_i/sigma_i where
sigma_i is the width of a uniform posterior over a smaller interval.
But later they say that they approximate tau_i by |w_i| + kappa_i with
kappa_i = \gamma |w_i| + epsilon.  I believe this works out to be
essentially a log-uniform prior on |w_i| (over some finite range of
log |w_i|).  This seems quite reasonable but should be made explicit.

The paper ignores the possibility that the prior should be centered at
the random initialization of the parameters.  This was found to be
essential in Dziugaite and Roy and completely changes the dependence
of k_i on w_i.

Another complaint is that the Hoefding bound is very loose in cases
where the emperical loss is small compared to its upper bound.  The
analysis can be more intuitively related to practice by avoiding the
rescaling of the loss into the interval [0,1] and writing expressions
in terms of a maximum bound on the loss L_max.  When hat{L} << L_max
(almost always the case in practice) the relative Chernoff bound is
much tighter and significantly alters the analysis.  See McAllester's
PAC-Bayesian tutorial.

The theoretical discussion on re-parameterization misses an important
point, in my opinoin, relative to the need to impose a learning bias
(the no-free-lunch theorem).  All L_2 generalization bounds can be
interpreted in terms of a Gaussian prior on the parameters.  In all
such cases the prior (the learning bias) is not invariant to
re-parameterization.  All L_2 generalization bounds are subject to the
same re-parameterization criticism.  A prior tied to a particular
parameterization is standard practice in machine learning for in all
L_2 generalization bounds, including SVMs.  I do think that a
log-uniform prior (rather than a Gaussian prior) is superior and
greatly reduces sensitivity to re-parameterization as noted by the
authors (extremely indirectly).

I did not find the empirical results to very useful.  The value of
parameter perturbation in training remains an open question. Although
it is rarely done in practice today, it is an important fundamental
question. A much more thorough investigation is needed before any
conclusions can be drawn with confidence. Experimentation with
perturbation methods would seem more informative than theory given the
current state of the art in relating theory to practice.

---

> ### Author Response · Authors · 2018-11-25
> **Thank you for your comments. We found that your comments gave us a different perspective on the problem.**
>
> Thank you for your comments. We found that your comments gave us a different perspective on the problem. For your questions:
>
>
>
> Q: I believe this works out to be
> essentially a log-uniform prior on |w_i| (over some finite range of
> log |w_i|).
> A: This looks like an interesting angle. However, we do not quite understand how the prior can be interpreted as log-uniform. The log term only comes from the KL divergence instead of the prior. Or maybe you are talking about a different interpretation using the Chernoff bound instead?
>
>
> Q: The paper ignores the possibility that the prior should be centered at
> the random initialization of the parameters.  This was found to be
> essential in Dziugaite and Roy and completely changes the dependence
> of k_i on w_i.
>
> A: Thanks for the great suggestion. We agree that there is no particular reason the prior should be zero centered. Trying priors with non-zero mean is definitely something to be investigated in future. Still one can get reasonable bound even with zero-centered priors. For example, Neyshabur et al. (https://arxiv.org/abs/1707.09564) also use priors with zero mean in their spectral norm bound for neural network.
>
>
> Q: Another complaint is that the Hoefding bound is very loose in cases
> where the emperical loss is small compared to its upper bound.  The
> analysis can be more intuitively related to practice by avoiding the
> rescaling of the loss into the interval [0,1] and writing expressions
> in terms of a maximum bound on the loss L_max.  When hat{L} << L_max
> (almost always the case in practice) the relative Chernoff bound is
> much tighter and significantly alters the analysis.  See McAllester's
> PAC-Bayesian tutorial.
>
> A: Thanks very much for the insightful suggestion. Actually Chernoff bound may lead to even cleaner solutions compared to the Hoeffding bound. We plan to investigate this further in future work.
>
>
> Q: The theoretical discussion on re-parameterization misses an important
> point, in my opinoin, relative to the need to impose a learning bias
> (the no-free-lunch theorem).  All L_2 generalization bounds can be
> interpreted in terms of a Gaussian prior on the parameters.  In all
> such cases the prior (the learning bias) is not invariant to
> re-parameterization.  All L_2 generalization bounds are subject to the
> same re-parameterization criticism.  A prior tied to a particular
> parameterization is standard practice in machine learning for in all
> L_2 generalization bounds, including SVMs.  I do think that a
> log-uniform prior (rather than a Gaussian prior) is superior and
> greatly reduces sensitivity to re-parameterization as noted by the
> authors (extremely indirectly).
>
> A: We agree. Originally, the PAC-Bayes bound talks about priors over the function space instead of the parameter space. In that case, the bound is invariant to reparameterization. Unfortunately, for ease of theoretical analysis, the prior is typically imposed on the parameter space which results in said issues.
>
>
> Q: I did not find the empirical results to very useful.  The value of
> parameter perturbation in training remains an open question. Although
> it is rarely done in practice today, it is an important fundamental
> question. A much more thorough investigation is needed before any
> conclusions can be drawn with confidence. Experimentation with
> perturbation methods would seem more informative than theory given the
> current state of the art in relating theory to practice.
>
> A: As suggested by you and reviewer 2, we have added some comparisons with the dropout method. In particular, we tried dropout with values 0.1, 0.3 and 0.5 and found that the perturbation analysis still yields better performance. We hope that this strengthens our claims about practical usefulness. We would be glad to add more experiments if you have any recommendations.

---

> > ### Comment · AnonReviewer1 · 2018-11-27
> > **the log-uniform thing and the re-parameterization thing.**
> >
> > My claim that you have an implicit log-uniform prior comes from the equation
> >
> > kappa(w) = gamma|w| + epsilon
> >
> > Here we are taking the width of the interval to be proportional to size of the parameter.  Under a log-uniform prior a similar choice of posterior gives a KL-divergence from posterior to prior that is independent of the size of w --- there is no KL divergence penalty for making w large.  In particular we could take the posterior to be the restriction of a log-uniform prior to a fixed interval in log-space. The width of the interval in log-space, and hence the KL-divergence from posterior to prior, is then independent of the size of w.
> >
> > I'm a little confused about your response on re-parameterization.  A typical Bayesian discussion of L_2 regularization would view the regularization as a log-prior term where the prior is a Gaussian prior on parameters (not functions).  This directly translates to PAC-Bayesian bounds under a Gaussian prior on parameters.  Re-parameterization changes the prior (the bias) if we use a coordinate-relative Gaussian prior.

---

> > > ### Author Response · Authors · 2018-11-28
> > > **thanks for the detailed explanation**
> > >
> > > Q: My claim that you have an implicit log-uniform prior comes from ...The width of the interval in log-space, and hence the KL-divergence from posterior to prior, is then independent of the size of w.
> > >
> > > A: Thanks for the detailed explanation. We agree, if the posterior is also log-uniform then dependency on w gets canceled out in the KL-divergence. Authors are unable to upload new manuscript PDFs at the moment so we will make this explicit in the final version.
> > >
> > >
> > > Q: I’m a little confused about your response on re-parameterization.  A typical ...
> > >
> > > A: Sorry for the confusion. Yes, the L2 regularization can be viewed as Gaussian prior over the parameters, and as you pointed out, the L2 regularizers are not invariant to reparameterization.
> > >
> > > In our last comment, we were saying that if possible, it is preferred to talk about distributions over the function space f: x-> y,  which may or may not be parameterized. In this way even if we do a reparameterization, the mapping f: x->y is considered the same, so it won’t affect the bound at all (ideally). Now for the ease of technical analysis, we parameterize the functions. In this way, the distribution over the functions spreads its mass across all the “equivalent” parameters.
> > >
> > > Still this is only a rough thought. There may be other technical difficulties on that, for example, if we avoid parameterization, how to define perturbation and so on.

---

### Official Review · AnonReviewer2 · 2018-10-22
**An extension of Neyshabur et al. PAC-Bayes bounds.**

**Rating:** 5
**Confidence:** 4

**Review:**

The authors prove a PAC-Bayes bound on a perturbed deterministic classifier in terms of the Lipschitz constant of the Hessian. They claim their bound suggests how insensitive the classifier is to perturbations in certain directions.

The authors also “extract” from the bound a complexity measure for a particular classifier, that depends on the local properties of the empirical risk surface: the diagonal entries of the Hessian, the smoothness parameter of the Hessian, and the radius of the ball being considered.  The authors call this “metric” “PAC-Bayes Generalization metric”, or pacGen.

Overall, this seems like a trivial extension of Neyshabur et al. PAC-Bayes bounds.

The experiments demonstrating that pacGen more or less tracks the generalization error of networks trained on MNIST dataset is not really surprising. Many quantities track the generalization error (see some of Bartlett’s, Srebro’s, Arora’s work). In fact, these other quantities track it more accurately. Based on Figure 2, it seems that pacGen only roughly follows the right “order” of networks generalizing better than others. If pacGen is somehow superior to other quantities, why not to evaluate the actual bound? Or why not to show that it at least tracks the generalization error better than other quantities?

The introduction is not only poorly written, but many of the statements are questionable. Par 2: What complexity are you talking about? What exactly is being contradicted by the empirical evidence that over-parametrized models generalize?

Regarding the comment in the introduction: “ Dinh et al later points out that most of the Hessian-based sharpness measures are problematic and cannot be applied directly to explain generalization.”, and regarding the whole Section 5, where the authors argue that their bound would not grow much due to reparametrization:
If one obtains a bound that depends on the “flatness” of the minima, the bound might still be useful for the networks obtained by SGD (or other algorithms used in practice). The fact that Dinh et al. paper demonstrates that one can artificially reparametrize and change the landscape of a specific classifier does not contradict any generalization bounds that rely on SGD finding flat minima. Dinh et al. did not show that SGD finds classifiers in a sharp(er) minima that generalize (better).

In the experiment section, the authors compare train and test errors of perturbed (where the perturbation is based on the Hessian) and unperturbed classifiers. However, they don't compare their results to other type of perturbations, e.g. dropout. It’s been shown in previous work that certain perturbations improve generalization and test error.

There are numerous typos throughout the paper.


****************

[UPDATE]

I would like to thank the authors for implementing the changes and adding a plot comparing their algorithm with dropout. While the quality of the paper has improved, I think that the connection between the perturbation level and the Hessian is quite obvious. While it is a contribution to make this connection rigorous, I believe that it is not enough for a publication. Therefore, I recommend a rejection and I hope that the authors will either extend their theoretical or empirical analysis before resubmitting to other venues.

---

> ### Author Response · Authors · 2018-11-25
> **We appreciate your suggestions and comments to make this draft better.**
>
> We appreciate your suggestions and comments to make this draft better.
>
>
> Q: Overall, this seems like a trivial extension of Neyshabur et al. PAC-Bayes bounds.
>
> A: While it is true that we are inspired by Neyshabur et al.’s great work, we respectfully disagree and emphasize that the extension is not trivial:
>
> 1. This is the first work connecting the local Hessian with the model generalization rigorously. There was no detailed quantitative analysis on how the Hessian and high-order smoothness terms could be integrated in the model generalization in Neyshabur et al.’s work.
>
> 2. Naively extending Neyshabur’s PAC-Bayes bounds wouldn’t give a closed-form solution for the perturbation levels since the bound they use has a square-root dependency on the KL-divergence. Instead, we handle the issue by starting from the PAC-Hoeffding bound with one additional dependency on \eta. The dependency on \eta is removed in Theorem 2 after the “optimal” perturbation level is solved by using the union bound over a grid of \eta.
>
>
> Q: Many quantities track the generalization error (see some of Bartlett’s, Srebro’s, Arora’s work).
>
> A: Yes, Barlett, Srebro, and Arora have amazing work to bound the generalization errors, but none of them explains how the local Hessian and smoothness terms can be related, in general, to model generalization.
>
>
> Q: In fact, these other quantities track it more accurately.  Or why not to show that it at least tracks the generalization error better than other quantities?
>
> A: These other quantities may or may not track the generalization more accurately depending on different scenarios.
> Bartlett’s and Srebro’s bounds contain the product of the norms of the coefficients, while our bound is more related to the local smoothness terms such as the neighborhood \kappa, Hessian and the Lipschitz constant of Hessian.
> An ideal case for our bound is when the loss function is flat over a large neighborhood around w^*. In this case the gap between the empirical loss and the population loss in our bound is about \sqrt{m/n} (ignoring the \log(1/\delta) term), but the gap in Bartlett and Srebro’s quantity could be large if the coefficient norm is large.
>
>
> Q: Based on Figure 2, it seems that pacGen only roughly follows the right “order” of networks generalizing better than others. If pacGen is somehow superior to other quantities, why not to evaluate the actual bound?
>
> A: Thanks for the great suggestion. Unfortunately it is difficult to calculate the bound exactly due to technical difficulties involved in the estimation of the Lipschitz constant of the Hessian in the local neighborhood. Instead, we propose heuristically approximated quantities like pacGen. As we mention in the last paragraph of Section 5, this algorithm and metric are not rigorous.
>
>
> Q: The introduction is not only poorly written, but many of the statements are questionable. Par 2: What complexity are you talking about? What exactly is being contradicted by the empirical evidence that over-parametrized models generalize?
>
> A: We apologize we did not make it clear in the introduction. The complexity we intended to discuss was the Rademacher complexity of the hypothesis space. The contradiction is as suggested by C. Zhang et al. that over-parameterized neural network can fit any function of sample size n, making the Rademacher complexity large, but empirically those neural networks generalize fairly well. We added a footnote to make this clear in the updated version. We have also made edits to the introduction to improve the flow and fix grammatical errors.
>
>
> Q: Regarding the comment in the introduction: “ Dinh et al ... reparametrization:
> If one obtains a bound that depends on the “flatness” of the minima, ... Dinh et al. did not show that SGD finds classifiers in a sharp(er) minima that generalize (better).
>
> A: Thanks for the comments. We agree.
>
>
> Q: In the experiment section, the authors compare train and test errors of perturbed (where the perturbation is based on the Hessian) and unperturbed classifiers. However, they don't compare their results to other type of perturbations, e.g. dropout. It’s been shown in previous work that certain perturbations improve generalization and test error.
>
> A: Thanks for the suggestion. We added the result using dropout in the appendix. We tried dropout rates of 0.1, 0.3 and 0.5. The results are plotted together with the perturbation algorithm. Note that dropout can be viewed as a multiplicative perturbation. In theory, there is no guarantee for which one would work better. But, empirically we observe better performance on the perturbed algorithm. One possible explanation is that dropout is set the same on all parameters, but the perturbation algorithm makes use of the local smoothness property for each parameter.

---

### Official Review · AnonReviewer3 · 2018-11-02
**Worth publishing work deserving a more rigorous presentation**

**Rating:** 6
**Confidence:** 4

**Review:**

The authors study generalization capabilities of neural networks local minimums thanks to a PAC-Bayesian analysis that grasps the local smoothness properties. Even if some assumptions are made along the way, their analysis provides a metric that gives insight on the accuracy of a solution, as well as an optimization algorithm. Both of these result show good empirical behavior.

However, despite my favorable opinion, I consider that the paper presentation lacks rigor at many levels. I hope that the criticism below will be addressed in an eventual manuscript.

It is confusing that Equations (4) and (9) defines slightly differently \sigma*_i(w*,\eta,\gamma). In particular, the former is not a function of \eta.

The toy experiment of Figure 1 is said to be self-explainable, which is only partly true. It is particularly disappointing because these results appear to be really insightful. The authors should state the complete model (in supplementary material if necessary). Also, I do not understand Figures (b)-(c)-(d): Why the samples do not seem to be at the same coordinates from one figure to the other? Why (d) shows predicted green labels, while the sample distribution of (b) has no green labels?

It is said to justify the perturbed optimization algorithm that Theorem 1 (based on Neyshabur et al. 2017) suggests minimizing a perturbed empirical loss. I think this is a weak argument for two reasons:
(1) This PAC-Bayes bounds is an upper bound on the perturbed generalization loss, not on the deterministic loss.
(2) The proposed optimization algorithm is based on Theorem 2 and Lemma 3, where the perturbed empirical loss does not appear directly.
That being said, this does not invalidate the method, but the algorithm justification deserves a better justification

There is a serious lack of rigor in the bibliography:
- Many peer-reviewed publications are cited just as arXiv preprints
- When present, there is no consistency in publication names. NIPS conference appears as "Advances in Neural ...,", 'NIPS'02", "Advances in Neural Information Processing Systems 29", "(Nips)". The same applies to other venues.
- Both first name initials and complete names are used
- McAllester 2003: In In COLT
- Seldin 2012: Incomplete reference

 Also, the citation style is inconsistent. For instance, the first page contains both "Din et al, (2007) later points out..." and "Dziugaite & Roy (2017) tries to optimize..."

Typos:
- Page 3: ...but KL(w*+u | \pi) => KL(w*+u || \pi)
- In this/our draft: Think to use another word if the paper is accepted
- Line below Equation (5): \nabla^2 L => \nabla L (linear term)
- it is straight-forward -> straightforward

---

> ### Author Response · Authors · 2018-11-25
> **Thanks very much for your careful reading and insightful suggestions.**
>
> Thanks very much for your careful reading and insightful suggestions.
>
>
> Q: “It is confusing that Equations (4) and (9) defines slightly differently \sigma*_i(w*,\eta,\gamma). In particular, the former is not a function of \eta. “
>
> A: We apologize for the confusion. You are right; the \sigma^* in (9) in Lemma 3 is a function of eta. Theorem 2 gets rid of the dependency on \eta by carefully choosing the scales of the \sigma^* and calculating the tail probability over a grid on \eta. We have fixed this in the manuscript.
>
>
> Q: The toy experiment of Figure 1 is said to be self-explainable, which is only partly true. It is particularly disappointing because these results appear to be really insightful. The authors should state the complete model (in supplementary material if necessary). Also, I do not understand Figures (b)-(c)-(d): Why the samples do not seem to be at the same coordinates from one figure to the other? Why (d) shows predicted green labels, while the sample distribution of (b) has no green labels?
>
> A: We agree that Figure 1 was poorly explained; thanks for pointing this out. To fix that, we have added a detailed commentary in the Appendix explaining the setup. We also fixed the coordinate issue in the updated version. And for your questions:
>
> 1. The model is a 5-layer MLP with 2 neurons in each hidden layer and sigmoid as activation. There are no bias terms in the linear layers, and the weights are shared across layers. For the 2-by-2 linear coefficient matrix, we treat two entries as constants and optimize the other 2 entries. In this way the whole model only has two parameters w_1 and w_2.
>
> 2. Figure (b) shows the input distribution constructed by thresholding a mixture of 3 Gaussians from the median. Figure (c) and (d) are plotted by predictions using the two parameters at the sharp minimum and flat minimum, respectively. The prediction using the flat minimum does not fit the labels so well as the sharp one. That’s why (d) only shows green labels. On the other hand, (d) also shows the predicted labels from the flat minimum has a simpler classification boundary compared to (c).
>
>
> Q: It is said to justify the perturbed optimization algorithm that Theorem 1 (based on Neyshabur et al. 2017) suggests minimizing a perturbed empirical loss. I think this is a weak argument for two reasons:
> (1) This PAC-Bayes bounds is an upper bound on the perturbed generalization loss, not on the deterministic loss.
> (2) The proposed optimization algorithm is based on Theorem 2 and Lemma 3, where the perturbed empirical loss does not appear directly.
> That being said, this does not invalidate the method, but the algorithm justification deserves a better justification
>
> A: These are great questions. In regards to the question of justification, we were inspired by the work of Dziugaite and Roy (https://arxiv.org/abs/1703.11008) which also optimizes the perturbed generalization bound and not the deterministic empirical loss as itself. We agree that this is somewhat dissatisfying but unfortunately, we can’t currently see a way around it given that we wish to analyze a general loss function. For specific instances, such as the max-margin loss, it may be possible to exploit the specific properties of the loss and obtain a deterministic bound, as you suggest, by extending the work of Neyshabur et al. (https://arxiv.org/abs/1707.09564).
>
>
> Q: There is a serious lack of rigor in the bibliography:
> - Many peer-reviewed publications are cited just as arXiv preprints
> - When present, there is no consistency in publication names. NIPS conference appears as "Advances in Neural ...,", 'NIPS'02", "Advances in Neural Information Processing Systems 29", "(Nips)". The same applies to other venues.
> - Both first name initials and complete names are used
> - McAllester 2003: In In COLT
> - Seldin 2012: Incomplete reference
>
>  Also, the citation style is inconsistent. For instance, the first page contains both "Din et al, (2007) later points out..." and "Dziugaite & Roy (2017) tries to optimize..."
>
> Typos:
> - Page 3: ...but KL(w*+u | \pi) => KL(w*+u || \pi)
> - In this/our draft: Think to use another word if the paper is accepted
> - Line below Equation (5): \nabla^2 L => \nabla L (linear term)
> - it is straight-forward -> straightforward
>
> A: Thanks for the careful reading and helpful suggestions. We fixed all the bibliography issues you mentioned as well as the typos. For the citation style, it seems that natbib automatically uses “A et al.” for papers with 3 or more authors, but “A & B” for papers with 2 authors.

---

### Meta-Review · Area_Chair1 · 2018-12-12
**ICLR 2019 decision**

**Confidence:** 4
**Recommendation:** Reject

**Metareview:**

This paper proposes a generalization metric depending on the Lipschitz of the Hessian.

Pros: Paper has some nice experiments correlating their Hessian based generalization metric with the generalization gap,

Cons: The paper does not compare its results with existing generalization bounds, as there is substantial work in the area now.  It is not clear whether existing generalization bounds do not capture this phenomenon with different batch sizes/learning rates, and the necessity of having and explicit dependence on the Lipschitz of the Hessian.

The bound by itself is also weak because of its dependence on number of parameters 'm'.

The paper is poorly written and all reviewers complain about its readability.

I suggest authors to address concerns of the reviewers before submitting again.